# The Landscapes of Gluten Regulatory Network in Elite Wheat Cultivars Contrasting in Gluten Strength

**DOI:** 10.3390/ijms24119447

**Published:** 2023-05-29

**Authors:** Jiajun Liu, Dongsheng Li, Peng Zhu, Shi Qiu, Kebing Yao, Yiqing Zhuang, Chen Chen, Guanqing Liu, Mingxing Wen, Rui Guo, Weicheng Yao, Yao Deng, Xueyi Shen, Tao Li

**Affiliations:** 1Zhenjiang Academy of Agricultural Sciences, Jiangsu Academy of Agricultural Sciences, Jurong 212400, China; jjliu0430@outlook.com (J.L.); lds0538@163.com (D.L.); zjykb@163.com (K.Y.); chen852235436@163.com (C.C.); wmxcell2007@163.com (M.W.); g869489301@126.com (R.G.); 15305295030@163.com (W.Y.); dengyao16@163.com (Y.D.); xueyi_shen@163.com (X.S.); 2Key Laboratory of Plant Functional Genomics of the Ministry of Education/Jiangsu Key Laboratory of Crop Genomics and Molecular Breeding/Collaborative Innovation of Modern Crops and Food Crops in Jiangsu/Jiangsu Key Laboratory of Crop Genetics and Physiology, College of Agriculture, Yangzhou University, Yangzhou 225009, China; zp15861386181@163.com (P.Z.); ljq12697@gmail.com (G.L.); 3Excellence and Innovation Center, Jiangsu Academy of Agricultural Sciences, Nanjing 210014, China; 20190054@jaas.ac.cn; 4Testing Center, Jiangsu Academy of Agricultural Science, Nanjing 210014, China; 20072804@jaas.ac.cn

**Keywords:** wheat, grain quality, full-length sequencing, weighted gene coexpression network analysis (WGCNA), multiscale embedded gene coexpression network analysis (MEGENA), gene set enrichment analysis (GSEA)

## Abstract

Yangmai-13 (YM13) is a wheat cultivar with weak gluten fractions. In contrast, Zhenmai-168 (ZM168) is an elite wheat cultivar known for its strong gluten fractions and has been widely used in a number of breeding programs. However, the genetic mechanisms underlying the gluten signatures of ZM168 remain largely unclear. To address this, we combined RNA-seq and PacBio full-length sequencing technology to unveil the potential mechanisms of ZM168 grain quality. A total of 44,709 transcripts were identified in Y13N (YM13 treated with nitrogen) and 51,942 transcripts in Z168N (ZM168 treated with nitrogen), including 28,016 and 28,626 novel isoforms in Y13N and Z168N, respectively. Five hundred and eighty-four differential alternative splicing (AS) events and 491 long noncoding RNAs (lncRNAs) were discovered. Incorporating the sodium-dodecyl-sulfate (SDS) sedimentation volume (SSV) trait, both weighted gene coexpression network analysis (WGCNA) and multiscale embedded gene coexpression network analysis (MEGENA) were employed for network construction and prediction of key drivers. Fifteen new candidates have emerged in association with SSV, including 4 transcription factors (TFs) and 11 transcripts that partake in the post-translational modification pathway. The transcriptome atlas provides new perspectives on wheat grain quality and would be beneficial for developing promising strategies for breeding programs.

## 1. Introduction

Wheat is one of the most important cereal crops all over the world and has been globally planted as a major food for human consumption [1]. One of the contributing factors to the widespread plantation of common wheat is the unique properties of wheat dough, which allow it to be processed into a wide range of baked products, greatly enriching human tastes. The end-use quality of wheat flour is largely dependent on the content and composition of wheat seed storage proteins (SSPs), which form the wheat gluten network after adding water to wheat flour [2,3]. Wheat gluten classically includes two fractions: the gliadins and the glutenins. The gliadins are monomeric proteins, including α-, β-, γ- and ω-gliadins, while the glutenins are composed of subunits assembled in polymers and can be further classified into two groups: high molecular weight (HMW) and low molecular weight (LMW) [2]. The HMW glutenin subunits (HMW-GSs) and LMW glutenin subunits (LMW-GSs) are responsible for wheat dough strength and elasticity, positively contributing approximately 40–70% of the overall wheat dough viscoelastic characteristics [4,5,6].

Genes encoding HMW-GSs are presented on the long arms of chromosomes 1A, 1B, and 1D, namely, *Glu-A1*, *Glu-B1*, and *Glu-D1* [7]. Each locus comprises two tightly linked genes encoding the larger x-type and the smaller y-type subunits, theoretically leading to six HMW-GSs: 1Ax, 1Ay, 1Bx, 1By, 1Dx, and 1Dy [2,6]. Studies focusing on HMW-GSs and their roles in wheat dough properties have been explored a lot [8,9,10,11,12,13,14]. LMW-GSs are encoded by genes at *Glu-3* loci (*Glu-A3*, *Glu-B3*, and *Glu-D3*) on the short arms of group 1 chromosomes, and each locus has multiple alleles [15,16]. Many studies have reported that LMW-GSs exert a positive effect on the rheological properties of wheat dough [17,18,19,20]. The *Gli-1* loci on the short arms of group 1 chromosomes encode β-, γ-, and ω-gliadins, and the *Gli-2* loci on the short arms of homoeologous group 6 chromosomes encode α-gliadins [21,22]. The sodium-dodecyl-sulfate (SDS) sedimentation volume (SSV), influenced by the gliadins and the glutenins and their fractions [23,24], is a key wheat flour quality trait, plays a fundamental role in determining end-use products, and can be employed as a crucial indicator for detecting quality and the quantity of gluten [25]. Moreover, SSV usually serves as an early-generation selection parameter in wheat quality breeding programs due to the simplicity and convenience of this measurement [26].

Despite the fact that the composition of gluten-related genes in common wheat has a significant influence on the end-use quality, fertilization, as one of the environmental factors, is usually taken into consideration for wheat quality improvement. Nitrogen (N) fertilizer is one of the most important and essential elements both for wheat grain protein content (PC) and grain gluten quality [27]. It has been reported that both HMW-GSs and LMW-GSs, as well as total PC, significantly increased with the increase in N applications [14]. γ-gliadins and HMW-GSs were differentially accumulated under N topdressing timing, while LMW-GSs were moderately increased [28]. Moreover, it has been proven that high-N treatment facilitates protein polymerization [29].

Zhenmai-168 (ZM168) and Yangmai-13 (YM13) are two widely cultivated bread wheat cultivars in the Middle and Lower Valleys of the Yangtze River in China. Notably, these two cultivars demonstrate distinct characteristics in terms of gluten strength. To investigate the genetic signatures that underlie gluten networks, PacBio single-molecule real-time (SMRT) sequencing and Illumina RNA sequencing (RNA-seq) were employed. ZM168, derived from a cross of Sumai-6/Yangmai-17G59, is an elite soft winter wheat cultivar with strong gluten, and it has been the most popular Chinese commercial wheat cultivar in the Middle and Lower Valleys of the Yangtze River with the total plantation areas up to 680,000 ha since 2008 (China Seed Association, http://202.127.42.47:6006/Home/BigDataIndex (accessed on 4 May 2023)). ZM168 has desirable agronomic and grain quality traits; hence, it has been used extensively as a core germplasm in many breeding programs. More than 50% of strong gluten cultivars grown in the Middle and Lower Valleys of the Yangtze River have ZM168 in their pedigree. However, the mechanisms of strong gluten formation in ZM168 remain largely unknown. The main goal of this study was to understand the genetic basis for the strong gluten trait in ZM168 and to identify grain quality-related genes.

## 2. Results

### 2.1. Nitrogen Treatment Increased SDS Sedimentation and Crude Protein Content

From the cross-section of the two wheat grains, it can be seen that YM13 has a whiter and more powdery texture, while ZM168 grains are keratinized (Figure 1a). The PC in both cultivars is more than 13% under nitrogen application, with even higher levels observed in Z168N (ZM168 treated with nitrogen) when compared to Y13N (YM13 treated with nitrogen). Y13 (YM13 without nitrogen fertilization) has the lowest PC content, with only 11.4%. After nitrogen application, both cultivars showed a significant but distinct increase in SSV and PC when compared to their respective controls (Figure 1b,c). Z168N has the highest SSV out of all the samples being compared. The SSV of Z168 (ZM168 without nitrogen fertilization), even without nitrogen feed, was remarkably higher than that in Y13N samples (Figure 1c), while PC in Y13N is unexpectedly more abundant than that in Z168 (Figure 1b).

### 2.2. Construction of High-Quality Isoforms

A total of 336,956 and 436,610 circular consensus (CCS) reads were identified in Y13N and Z168N using the PacBio single molecular real-time (SMRT) sequencing platform, respectively (Appendix A). Of these CCS reads, a significant proportion was characterized as full-length non-chimeric (FLNC) reads, with 241,555 (71.69%) for Y13N and 370,087 (84.76%) for Z168N. These FLNC reads were further processed to obtain consensus isoforms. A total of 65,107 consensus isoforms were obtained for Y13N, of which 65,098 were deemed high-quality. After removing redundant sequences, 44,709 non-redundant transcripts were obtained. For Z168N, a total of 93,881 consensus isoforms were obtained, of which 93,863 were of high quality, and 51,942 were non-redundant transcripts. The data also indicates that a total of 31,647 and 38,614 genes were detected in Y13N and Z168N, respectively. This includes a large number of novel genes, with 6893 and 10,788 in Y13N and Z168N, respectively, as well as novel isoforms, with 28,016 and 28,626 in Y13N and Z168N, respectively (Appendix A).

### 2.3. Functional Annotations for Identified Isoforms

All non-redundant transcripts were annotated by searching against several databases, including NCBI non-redundant protein sequences (NR database), protein family (Pfam), clusters of orthologous groups of proteins (KOG/COG/eggNOG), protein sequence database (UniProt), kyoto encyclopedia of genes and genomes (KEGG) and gene ontology (GO). A total of 42,260 isoforms were well annotated, of which 32,410 (76.69%) and 15,141 (35.83%) transcripts were annotated with GO terms and KEGG pathways, respectively. To identify genes significantly enriched in GO terms, KEGG pathways, and Pfam, we employed gene set enrichment analysis (GSEA) for enrichment analysis. Overrepresented GO terms showed distinct responses to nitrogen treatment and differences between two cultivars. Accordingly, genes involved in chaperone-mediated protein complex assembly (GO:0051131), protein complex oligomerization (GO:0051259), protein homooligomerization (GO:0051260), and protein refolding (GO:0042026), were more abundant in Z168N when compared to Z168 and Y13N (Appendix A). GSEA also showed that some terms of molecular function were more enriched in Z168N than in its counterparts, Z168 and Y13N. These terms comprise functions in chaperone binding (GO:0051087) and heat shock protein (HSP) 90 protein binding (GO:0051879). Compared with the control Y13, protein homooligomerization (GO:0051260) and protein complex oligomerization (GO:0051259) are negatively correlated in Y13N (Appendix A). Pfam analysis suggested that transcripts belonging to the HSP20/alpha crystallin family (PF00011), HSP70 (PF00012) and HSP90 (PF00183) were more abundant in Z168N when compared with those in Y13N and Z168 (Appendix A). HSP-related proteins are annotated by KEGG analysis to partake in “protein processing in endoplasmic reticulum”, and this pathway also contains transcripts with DNAJ domain (PF00226), and DNAJ C terminal domain (PF01556). The DNAJ domain was significantly increased in abundance in the Z168N when compared with that in the Y13N, while not significant when compared with that in the Z168 (Appendix A). Genes with the DNAJ C terminal domain exhibited higher expression intensity in Z168N when compared with those in Z168 and Y13N (Appendix A). Transcripts in the cys-rich gliadin N-terminal (PF13016) family differed significantly after nitrogen treatment in both cultivars (Appendix A).

### 2.4. Isoforms Differentially Expressed under Nitrogen Treatment

A total of 829 transcripts were differentially expressed in Z168N versus Z168, and 457 transcripts in Y13N versus Y13. Among these, 45 transcripts were found to be common between the two groups (Figure 2a). The abundance of two gamma-gliadin transcripts was two times higher in Z168N than in Z168. The expressions of two alpha-gliadin transcripts dramatically soared to 24 and 496 times higher in Z168N than in Z168. Three omega gliadin transcripts were also upregulated in Z168N when compared with Z168. A total of 9177 DETs were found between Z168N and Y13N (Figure 2a), including 248 DETs that are shared between Y13N vs. Y13 and Z168N vs. Y13N, and 495 DETs that are shared between Z168N vs. Z168 and Y13N vs. Y13. Among these, a total of 28 DETs were common across all the three comparisons. A total of 63 Pfam were significantly enriched for DETs in Z168N versus Y13N (Figure 2b, Appendix A). Gluten-related genes belonging to the protease inhibitor/seed storage/LTP family (PF00234) and Cys-rich gliadin N-terminal (PF13016) exhibited higher levels of expression in Z168N when compared with Y13N (Appendix A). Genes in heat shock protein (HSP) families, including HSP20, HSP70, and HSP90, are significantly upregulated in Z168N as compared to those in Y13N (Appendix A). Predicted KEGG pathways relating to chaperones, folding catalysts, and protein processing in the endoplasmic reticulum were significantly enriched in Z168N when compared with Y13N (Figure 2c, Appendix A). GO terms (Figure 2d,e), including protein folding (GO:0006457), unfolded protein binding (GO:0051082), response to heat (GO:0009408), protein complex oligomerization (GO:0051259), protein homooligomerization (GO:0051260), and nutrient reservoir activity (GO:0045735), were significantly enriched in Z168N when compared with Y13N (Appendix A).

### 2.5. Co-Expression Network for SSV

To further explore potential connections between gene expression patterns and SSV, both weighted gene co-expression network analysis (WGCNA) and multiscale embedded gene co-expression network analysis (MEGENA) were performed to characterize specific modules. A total of 14 colored modules were constructed using WGCNA (Figure 3a,b), of which the brown module was considered the key module associated with SSV (correlation coefficient = 0.93, *p*-value = 2 × 10^−5^), and therefore it was further analyzed in detail (Appendix A). Gene significance and module membership were compared to investigate the correlations between individual genes and the brown module (Figure 3c). A total of 123 modules were constructed by MEGENA, of which 15 modules were significantly correlated with SSV (Figure 3d–f) under the strict screening threshold of a correlation coefficient > 0.8 and a *p*-value < 0.01 (Appendix A). Of the genes included in significant MEGENA modules, 35% also overlapped with the genes in the WGCNA brown module, and 73% of hub genes assigned by MEGENA fell into the WGCNA brown module (Appendix A). GSEA analysis demonstrated that mostly enriched GO terms and Pfam were shared between the WGCNA brown module and significant MEGENA modules (Appendix A).

### 2.6. Alternative Splicing Events in Significant Modules

Alternative splicing (AS) events were identified using Astalavista from all transcripts, and five types of AS were found: skipped exon (SE), mutually exclusive exon (MXE), alternative 5′ splice site (A5SS), alternative 3′ splice site (A3SS), and retained intron (RI). A total of 46,497 and 51,942 AS events were detected in Y13N and Z168N, respectively, with 12,430 events being shared between the two sets. The analysis resulted in the identification of 584 differential AS (DAS) events between Z168N and Y13N (Figure 4c,d), where SE was the predominant type, accounting for 51.54% (301) of all DAS, followed by A3SS (16.44%), RI (15.24%), A5SS (14.38%), and MXE (2.40%). Moreover, a total of 130 DAS events were found in the WGCNA brown module and 132 in the MEGENA significant module. Of these, 38 DAS events were shared, suggesting that these events may be essential and conserved in network regulation.

### 2.7. Identification of TFs, lncRNAs, and APA

Transcription factors (TFs) and transcription regulators (TRs) were predicted by iTAK software (version 1.7). A total of 17,762 isoforms of 13,487 genes from 48 TF families and 24 TR families were identified (Figure 4a). Compared with Z168, 39 TFs were up-regulated in Z168N, while 46 TFs were down-regulated. Rcd1-like, mTERF, and IWS1 were up-regulated in Z168N (compared with Z168) and down-regulated in Y13N (compared with Y13). HSF, FAR1, HB-HD-ZIP, and MBF1 were up-regulated in Z168N compared with Y13N (Figure 4b).

Four tools were employed to identify sequences without coding potential: CNCI, CPC, pfam, and CPAT. A total of 572 lncRNAs were predicted in common by all four methods. Four types of lncRNAs were identified, including lincRNA (510, 89.2%), antisense-lncRNA (14, 2.4%), intronic-lncRNA (3, 0.5%), and sense-lncRNA (45, 7.9%). The lncRNA targets were also predicted based on the location of lncRNAs and complementary base pairing with mRNA, 491 out of 572 lncRNAs have at least one target (Appendix A). The WGCNA brown module contains three lncRNAs, while the MEGENA significant modules contain six lncRNAs, with only one lncRNA being shared between them. The shared lncRNA was predicted to target eight genes, two of which were also identified in the WGCNA brown module (Appendix A). Additionally, one of the lncRNAs, PB.9183.1, was previously assigned to the c5_454 MEGENA module, and one of its potential targets was also included in the same module (Appendix A).

Polyadenylation of the 3′ end of mRNA can result in various gene expressions due to the generation of different transcripts from different 3′ untranslated regions. As a result, polyadenylation (APA) is a crucial post-transcriptional regulatory modification in plant development. The results found that 11,088 genes have at least one identified poly(A) site, with 9350 (84.3%) genes having a single poly(A) site. The average number of aligned reads per gene was approximately 9 and 10, yielding 9711 and 9713 unique ploy(A) sites for Y13N and Z168N, respectively. A MEME analysis was performed to identify enriched motifs 50 nucleotides upstream from the poly(A) sites of all transcripts [30]. The results showed that Poly(A) signals TGBTTG, KTGBTG, and CCGGCC were significantly over-represented.

## 3. Discussion

### 3.1. High-Quality Data Obtained from PacBio Transcriptome Sequencing

Third-generation sequencing technology holds great potential for improved transcript detection compared to second-generation sequencing techniques such as RNA-seq. This has been demonstrated through its successful applications in a variety of plants [31,32,33,34]. The current study provided comprehensive landscapes of isoform full-length sequences, gene structures, lncRNAs, and AS events. The transcript isolation and characterization were performed efficiently, utilizing the benefits of the PacBio platform [35]. The novel isoforms offered by PacBio sequencing contribute to common wheat genome annotation as well as gene cloning in wheat grain quality research. Previous studies have shown that AS events are frequently observed across multiple species [36,37,38,39,40]. However, the impact of AS on the transcriptional regulation during the formation of the wheat gluten fraction remains ambiguous. Discovered AS events in our results proposed that the coordination of AS with transcriptional regulation may contribute to elevated gluten fraction levels. Although the long reads produced by PacBio technology are superior to the second-generation sequencing; higher error rates of sequencing exist in third-generation sequencing [41]. To address this challenge, third-generation sequencing was performed in conjunction with second-generation sequencing, which is referred to as hybrid sequencing, and was utilized for further corrections [42]. On the other hand, third-generation sequencing has also been proposed to alleviate the ambiguities that currently exist in second-generation sequencing. Consequently, the hybrid analysis used in this study simultaneously detected distinct isoforms and provided valuable insights into the regulation of isoform abundances.

### 3.2. The Key Modules Show Strong Associations with SSV and Are Well-Constructed and Annotated

To derive a holistic picture of gluten regulatory network structures, we used both the traditional WGCNA and the new MEGENA method to systematically uncover key regulators. The WGCNA is a powerful tool for mining gene expression data, enabling the correlation of well-separated modules with phenotypic traits to identify intramodular hub genes. However, the limitation of this method is that one gene can be solely assigned to one module. Different from the simple gene membership in WGCNA, the MEGENA can generate a more comprehensive network, as one gene can be assigned to multiple modules in a multiscale approach. Only isoforms shared by significant modules from WGCNA and MEGENA were retained as key regulators. It is important to note that the criteria for defining significant MEGENA modules in this study were subjectively stringent, which may result in a lack of information about the network structures. For instance, the c5_471 module (see Appendix A for a detailed list) has a correlation coefficient of 0.79 and a *p*-value of 0.002, which suggests a potential association with SSV, but it was not considered a significant module.

The reliability of the multiscale network constructed herein was further supported by the results from a GSEA perspective. Functional annotations from GO and KEGG analysis of DETs and significant modules simultaneously demonstrated genes in Z168N were highly enriched for GO terms associated with protein folding and KEGG terms relating to protein processing pathways (see Appendix A for a complete list). Gluten fraction-related genes that are annotated with molecular function and nutrient reservoir activity (GO:0045735) were significantly upregulated in Z168N. In addition to GO and KEGG analysis, Pfam annotations were specifically included for identifying enriched protein families or domains, leading to a better understanding of specific activities during grain development. Over-represented Pfam terms reflected heat shock proteins that were actively upregulated in Z168N as compared to Y13N (Figure 5), including HSP20 (PF00011), HSP70 (PF00012), HSP90 (PF00183), and HSF (PF00447). Genes in the Glutenin_HMW (PF03157) clan did not exhibit significance in abundance as the two cultivars have different HWM subunits, where YM13 carries 2Dx and 12Dy subunits and ZM168 carries 5Dx and 10Dy subunits (unpublished). The wheat gliadins include two superfamily members: the gliadin domain (PF13016) and the tryp-alpha-amyl domain (PF00234) [43]. The gliadin domain superfamily comprises cysteine-rich gliadin N-terminal and avenin plant proteins, and the tryp-alpha-amyl domain is classified as a protease inhibitor/seed storage/LTP family and includes lipid transfer proteins (LTPs). The GSEA results showed that the PF13016 superfamily was significantly enriched in Z168N when compared to that in Z168; however, it was not significantly pronounced in Z168N compared to Y13N. On the other hand, it was found to be significant in Y13N compared to Y13. These results suggest that transcripts belonging to the gliadin domain are generally regulated by nitrogen application in both cultivars, which aligns with previous research findings [28]. On the contrary, the abundances of transcripts, which belong to the tryp-alpha-amyl domain superfamily, were significantly increased in Z168N when compared to both Z168 and Y13N, and there was no significant difference between Y13N and Y13, suggesting that these transcripts might play a crucial role in strong gluten fraction formation.

### 3.3. Hub Genes Involved in Regulatory of Strong Gluten Aggregates

Among these candidates (Figure 5, see Appendix A for a complete list), a total of 11 genes were annotated with potential functions in post-translational modification, protein turnover, and chaperones, and another three genes were involved in transcription. HSP70 [44] and HSP90 [45] are ubiquitous chaperone proteins, performing diverse functions in lots of species, especially in cellular protein folding [46]. It has been historically reported that HSPs are strongly upregulated by heat stress [47,48], and linked with other multiple stress responses [49]. The Hsp70/Hsp90 chaperone machinery in gluten aggregate formation remains largely unknown. A novel quantitative trait locus (QTL) on chromosome 1A (*QSsv.cau-1A.1*) is reported to be highly responsible for SSV trait variation, and it harbors one HSP70 gene (*TraesCS1A02G295600.1*) [50], indicating that the HSP70 gene might be a candidate for this locus. Similarly, another candidate gene HSP90 (*PB.13441.3*) was included in the *QTL-2D-1* interval, which is reported to be correlated with SSV, wet gluten content, grain protein content and dough water absorption [51], which were previously reported to be significantly correlated with SSV [52]. Additionally, some HSP70 genes were identified as one of the candidate genes for QTLs on chromosome 6D and 7B, which were documented to be associated with baking value and grain hardness properties [53]. Moreover, the QTL on chromosome 6D were also found to be controlling SSV, and three HSP70 genes fall into this QTL region. Another QTL on chromosome 2B was predicted to be related to gluten content, as it contains one HSP gene. Hence, substantial interest has arisen in deciphering the HSP gene regulatory network responsible for SSV trait variation. Furthermore, the HSP70/HSP90 co-chaperone HOP (HSP70-HSP90 organizing protein) was reported to bind to both Hsp70 and Hsp90 at the same time and to assist the delivery of client peptides from HSP70 to HSP90 [54]. Another gene that was documented to be involved in HSP70/HSP90 regulatory network is BAG family molecular chaperone regulator 4 (*BAG4*), which mediates as a nucleotide exchange factor to promote protein binding and aggregating that was prevented by HSP70 due to partially incomplete syntheses [55]. Expression levels of two HSP90 genes (*TraesCS2A02G033700.1*, *PB.13441.3*), three HSP70 genes (*PB.16957.1*, *TraesCS1D02G131800.1*, *TraesCS1A02G295600.1*), the HOP gene (*TraesCS2A02G386800.1*) and the *BAG4* gene (*PB.20810.1*) were dramatically elevated in Z168N when compared with those in Y13N, and in Z168 when compared to those in Y13, suggesting their potential involvement in strong gluten fraction formation.

In addition to HSP-related gene controlling the synthesis of strong gluten in wheat grain, several TFs were found to be key regulators for the formation of gluten fraction in wheat. Some TFs have been previously characterized to be involved in wheat storage protein expression manipulation, such as wheat prolamin-box binding factor (*WPBF*) [56], wheat storage protein activator (*TaSPA*) [57] and NAM/ATAF/CUC (NAC) transcription factor 19 (*TaNAC019*) [58]. Some other TFs were found to be candidate genes for QTLs on the chromosome 4B and 6B, and were previously reported to be associated with SSV [53]. In the work presented herein, we found that an HSF gene was markedly upregulated in Z168N when compared to that in Y13N, and both isoforms (*PB.2006.2* and *PB.2006.4*) for this gene were identified as hub players in MEGENA regulatory network, suggesting that this gene may play a positive role in regulating wheat gluten fractions. Similar results were observed for the transcription initiation factor IIF, alpha subunit (*TFIIF-alpha*, *PB.25547.2*), which was differentially expressed between Z168N and Y13N, and it differs from previous report of their potential roles in wheat seed hardness [53] and thousand grain weight [59]. Thus, in addition to the putative functions reported, *TFIIF-alpha* may have activities in the regulation of wheat grain qualities.

## 4. Materials and Methods

### 4.1. Plant Materials and Nitrogen Treatment

Two cultivars were used as the materials, including ZM168 and YM13, and they were normally grown at the experimental field stationed at Zhenjiang Agricultural Research Centre. A total of 12 plots (each plot is composed of 20 rows spaced 20 cm apart, with each row measuring 1.5 m in length) were prepared in the field, with thirty seeds in each row. For each cultivar, three plots were treated equally using side dressing with 250 g of 46% carbamide fertilizer at the stem elongation stage and then at heading developmental stage. Therefore, four samples were included: Y13 and Z168 (the controls without nitrogen fertilization), Y13N and Z168N (treated with nitrogen). A total of 30 spikes were randomly selected from different plants, and only one grain from the middle spikelet of each spike was collected 20 days post flowering for RNA extraction, as the wheat grain already started dough development, with continues filling carbohydrate and protein at this time point. All three biological replicates were included for each sample.

### 4.2. Qualitative Traits Measurement

Mature seeds were harvested for measuring SSV and crude PC. The PC was ascertained by Perten DA 7250-NIR analyser (Perkin Elmer, Waltham, MA, USA, AACCI Approved Method 39–11.01). The content and quality of gluten proteins can be evaluated through SSV, which has a considerable effect on end-use quality of wheat, including flour strength, elasticity, extensibility and cohesiveness. The measurement procedure adhered to the National Standard of China GB/T 15685-2011 [60] with slight alterations. Briefly, 2 g flour were weighed and mixed with 30 mL water in a glass-cover measuring cylinder and were shaken for 5 min with a shaker, then followed by 30 mL pre-mix solution (2% SDS with 20 mL 1.2 mol/L lactic acid) and after being shaken for 5 min, the flocky precipitate volume was recorded after 30 min silence. Three duplicates were carried out for each sample.

### 4.3. PacBio Full-Length Data Processing and Illumina RNA Sequencing

The total RNA of four samples was used for Illumina RNA sequencing [61], and the RNA from three replicates of Z168N and Y13N was bulked for PacBio sequencing, respectively [35]. The circular consensus (CCS) sequences were obtained from raw reads by conducting Iso-seq pipeline with minFullPass = 3 and minPredictedAccuracy = 0.9. Then full-length, non-chemiric (FLNC) transcripts were determined from CCS as it contains complete polyA tail signal and the 5′ and 3′ cDNA primers. SMRTLink (version 11.0) was used for obtaining consensus isoforms. Unavoidably, as the existence of the degradation on 5′ end, gene optimization was necessarily conducted by removing the redundant sequences before the identification of high-quality transcripts. Full-length consensus sequences were mapped to IWGSC genome reference sequence of Chinese Spring (RefSeqv1.1) [62] using Genomic Mapping and Alignment Program (GMAP, http://research-pub.gene.com/gmap/ (accessed on 4 May 2023)) [63]. To assess the completeness of the whole genome assembly, the transcriptome data and Benchmarking Universal Single Copy Orthologs (BUSCO, version 5.3.2) [64] were employed for evaluation. Additionally, BUSCO was run in transcriptome mode, and genome sequences were searched against the lineage dataset: embryophyta_odb9 (creation date: 2016-02-13, number of species: 30, number of BUSCOs: 1440).

For RNA-Seq analysis, clean data were obtained from raw reads by removing reads containing adapter and ploy-N and reads with low quality were discarded. The Q20, Q30, GC-content, and sequence duplication levels of the clean data were calculated. After clean data were guaranteed to be of high quality, they were then aligned to the reference genome sequence using Hisat2 tool (version 2.2.0) [65]. The quantification of assembled transcript expression (Transcripts Per Kilobase Million, TPM) was conducted using StringTie (version 2.1.5) [65]. Differential expression was analyzed with DESeq2 (version 1.28.1) [66] based on raw transcript counts, and the differentially expressed transcripts (DETs) were screened with an adjusted *p*-value < 0.01 and |log2 fold change| ≥ 1.

### 4.4. Structure Analysis

Five types of AS events were previously observed: exon skipping (ES), mutually exclusive exons (MXE), intron retention (IR), alternative 5′ donor site (A5SS), and alternative 3′ acceptor site (A3SS) [67]. All AS events were identified by the AStalavista tool. DAS events were detected by Multivariate Analysis of Transcript Splicing (MATS, version 3.2.1) [68], which is a computational tool for calculating *p*-value and false discovery rate (FDR) from RNA-Seq data where the differences occurred between isoforms of a certain gene under a comparison. The coding potential calculator (CPC) [69], coding-non-coding index (CNCI) [70], coding potential evaluation tool (CPAT) [71], and protein domain [72] analysis were combined to screen transcripts without coding potential, namely long non-coding RNAs (lncRNAs). For identification and classification of different types of TFs, iTAK was conducted based on the protocols from Zheng, et al. [73]. Alternative polyadenylation (APA) site analysis was implemented with TAPIS [74] over FLNC, and novel motifs were predicted by MEME suite [75].

### 4.5. Functional Annotation and Enrichment Analysis

Isoform function was annotated based on the following databases: NR database [76], Pfam [77], KOG/COG/eggNOG [78,79], UniProt [80], KEGG [81], and GO [82]. GO, KEGG, and Pfam enrichment analyses for whole transcripts were assessed by GSEA [83], which systematically highlights enriched gene sets by ranking differentially expressed genes between two samples. The expression matrix was subjected to GSEA as the “expression dataset” for further analysis. Parameters were set up as follows: number of permutations: 1000, enrichment statistics: weighted, metric for ranking genes: Signal2Noise. Terms with *p* < 0.05 and FDR < 0.05 were assigned as significant.

### 4.6. WGCNA and MEGENA

In order to depict a comprehensive and effective regulatory network, DETs were characterized at the threshold of *p*-value < 0.05, and meanwhile, quantified transcripts were only considered to be expressed with TPM > 0.5 in at least two samples. After gene expression matrix was prepared, two methods were applied to puzzle out tissue-specific co-expression networks: WGCNA (version 1.70-3) and MEGENA (version 1.3.7). WGCNA is normally used to find phenotype-related modules and hub genes among the whole expression profile [84]. We set the soft-threshold power = 9, cut height = 0.25, corType = “Pearson” and networkType = “unsigned” for network construction and module detection. The module with the highest correlation with SSV was considered as the key module, in which transcripts were served as the hub isoforms/genes. Gene significance (GS) and a module membership (MM) were evaluated for hub genes in the key module. Different from WGCNA, the newly developed MEGENA method can assign a certain gene into multiple modules at multiple scales, leading to large number of modules with smaller sizes [85]. Parameters for MEGENA were set as follows: minimum module size = 50, method for correlation = “spearman”, FDR cut-off = 0.05, number of permutations calculating connectivity significance *p*-value = 100. The gene expression patterns were displayed by using “TCseq” (version 1.12.1) in R. 

### 4.7. RT-PCR

Reverse-transcription PCR (RT-PCR) was conducted to assess the consistency between MATS results and their corresponding DAS events. Total RNA was extracted using FastPure total RNA isolation kit (RC411, Vazyme Biotech Co., Ltd. (Nanjing, China)) according to the manufacturer’s protocol. RT-PCR was performed using bulked RNA from exactly the same samples prepared for PacBio sequencing to maintain consistency. RNA was reverse transcribed using Takara (Takara Biomedical Technology, Beijing, China) PrimeScript RT reagent kit. The primers for each gene were designed by Primer Premier 6.0. Same PCR reactions were used for all thermal profiles: 95 °C for 5 min, followed by 35 cycles at 95 °C for 20 s, 55 °C for 30 s, 72 °C for 45 s, and then 72 °C for 5 min. PCR products were separated into a 1.5% agarose gel.

## 5. Conclusions

In summary, our multiscale regulatory network, together with GSEA analysis has systematically unveiled a number of key regulators associated with the SSV trait. We annotated 11 genes with post-translational modification functions and 4 genes involved in transcription, as well as the differential expression of previously reported wheat grain genes in strong and weak gluten cultivars. Our findings provide contrasting information between ZM168 and YM13, which will aid in QTL mapping and gene cloning. However, our integrative network also revealed additional, under-explored regulators. Further dissection of HSP genes in wheat gluten fractions is needed to fully understand the wheat gluten architecture.

## Figures and Tables

**Figure 1 ijms-24-09447-f001:**
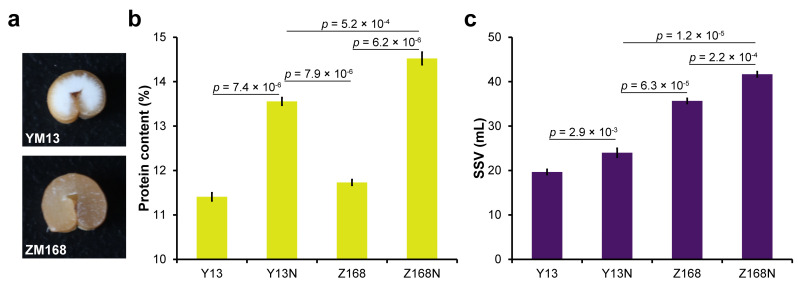
(**a**) Mature seed cross-section of the ZM168 and YM13 cultivars. (**b**) Quantification of protein content (PC) in four samples. (**c**) Sodium-dodecyl-sulfate (SDS) sedimentation volume (SSV) in four samples. Two-tailed Student’s *t* test was employed for group comparison. All data are presented as mean ± SD.

**Figure 2 ijms-24-09447-f002:**
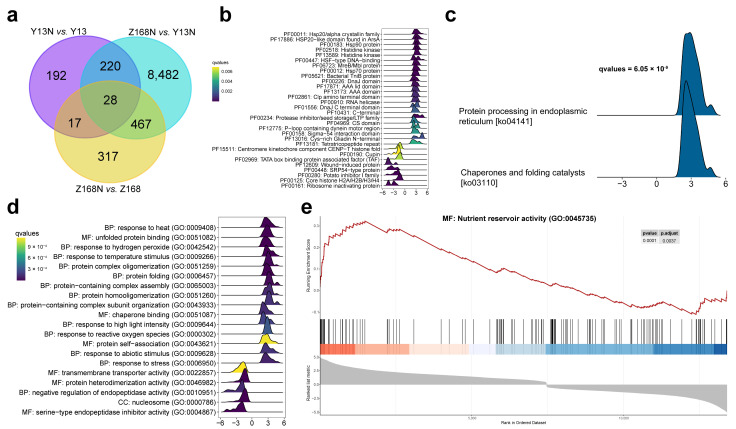
Differentially expressed transcripts (DETs) and their functional enrichment analysis. (**a**) The number of DETs under three pairs of comparison: Y13N versus Y13, Z168N versus Z168, and Z168N versus Y13N. DETs were screened by log2 fold-change ≥ 1 or ≤ −1 and adjusted *p*-value ≤ 0.01. (**b**) Top 30 significant Pfam terms in Z168N versus Y13N. The plot displays the enrichment score (ES) as a function of the position of the genes in the ranked list, with positive scores indicating enrichment in Z168N and negative scores indicating enrichment in Y13N. Colors in the ridgeline plot represent respective q-values. (**c**) Significantly enriched KEGG pathways in Z168N with an ES of 0.49 and 0.51 for chaperones and folding catalysts pathway and protein processing in endoplasmic reticulum pathway, respectively. (**d**) Top 20 significantly enriched gene ontology (GO) terms in Z168N versus Y13N. (**e**) Gene set enrichment analysis (GSEA) plot for isoforms annotated with nutrient reservoir activity function, normalized enrichment score = 1.89. The red line indicates the running sum of the ES, with the peaks corresponding to gene sets that are significantly enriched in Z168N. GSEA was used for all enrichment analyses, and detailed results are provided in Appendix A.

**Figure 3 ijms-24-09447-f003:**
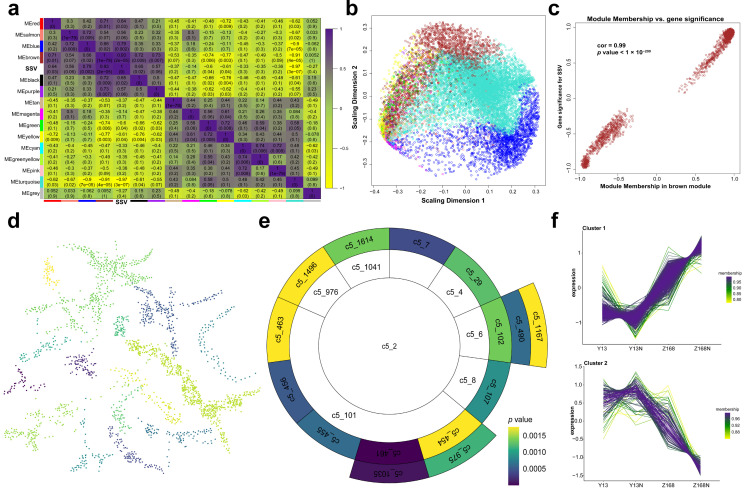
Regulatory network construction and module-trait relationship identification. (**a**) Associations between SSV and the modules in WGCNA. Each cell in the heat map contains the corresponding correlation coefficient (up) and *p*-value (down). The brown module has the highest correlation with SSV, where correlation coefficient = 0.93 and *p*-value = 2 × 10^−5^. (**b**) Multidimensional scaling (MDS) plot for WGCNA modules. (**c**) Scatterplot of gene significance versus module membership in brown module. Genes were separated into two distinct groups: genes plotted on top right are positively correlated with SSV in Z168N, while genes plotted on the bottom left displayed significant correlations with SSV in Y13N. (**d**) The whole network of significant modules identified by MEGENA, nodes that share the same cluster were colored the same. Significant modules were screened by a correlation coefficient ≥ 0.8 and *p*-value ≤ 0.01. Colors corresponded to *p*-values. (**e**) Sunburst for significant modules in MEGENA. Blank modules corresponded to each parent module. See Appendix A for the complete list of the module-trait relationships. (**f**) Expression patterns of genes in MEGENA-significant modules. It was generally separated into two groups, the top one displays that genes were substantially upregulated in Z168 and Z168N compared with Y13 and Y13N, and even more abundant in Z168N. The other one depicted the opposite expression patterns.

**Figure 4 ijms-24-09447-f004:**
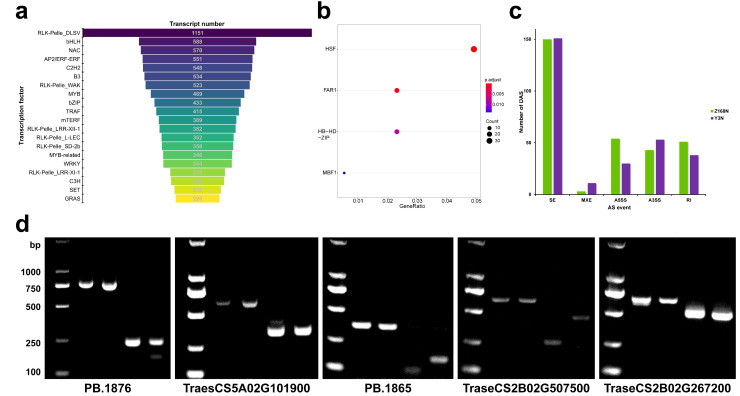
(**a**) The number of top 20 transcription factors (TFs). (**b**) Significantly enriched TFs in Z168N versus Y13N. The FDR for HSF, FAR1, HB-HD-ZIP and MBF1 were 2.43 × 10^−17^, 6.87 × 10^−4^, 7.1 × 10^−3^ and 0.012, respectively. (**c**) The number of differential alternative splicing (DAS) events in Z168N versus Y13N. Number of significant events detected using junction counts only. (**d**) PCR-based verification for randomly selected DAS events. Samples from left to right are genomic DNA of YM13, genomic DNA of ZM168, Y13N cDNA, and Z168N cDNA.

**Figure 5 ijms-24-09447-f005:**
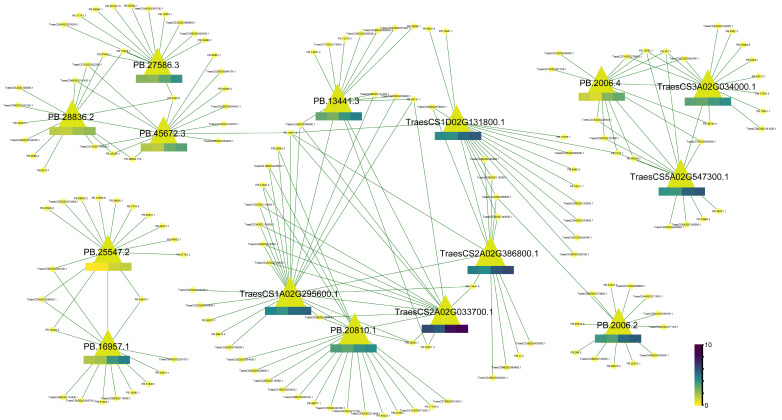
The network of hub genes identified by MEGENA. The 15 candidates are represented by yellow-green triangle, while other hub genes in significant MEGENA modules are represented in purple circle. The heatmap for each candidate gene is displayed below its name, with square colors indicating the expression levels on a log2(TPM + 1) scale across four samples (from left to right): Y13, Y13N, Z168, and Z168N.

## Data Availability

Not applicable.

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
