# Peer review of "The Landscapes of Gluten Regulatory Network in Elite Wheat Cultivars Contrasting in Gluten Strength"

_ijms, 2023, doi:10.3390/ijms24119447_

Round 1

Reviewer 1 Report

Manuscript The landscapes of gluten regulatory network in elite wheat cultivars contrasting in gluten strength compares two wheat cultivars, with strong and weak gluten using PacBio and RNA-seq analysis results via bioinformatical computiations. Overall, the study is of high quality level and corresponds to the audience of IJMS.

However, there are several issues that should be corrected and/or addressed 

Results

line 134-135  - GO:0061077 and GO:0006457 not found in Supplementary Table S3, therefore, they cannot be mentioned in the senence where Z168N vs. Z168 are compared. 

lines 141-142 - GO:0051260 and GO:0051259 not found in Supplementary Table S4, while the sentence has reference to it. 

lines 144-145 - "were upregulated in Z168 when compared with Y13" - there is no comparison between Z168 and Y13 in the present study. 

line 146 - "(Supplementary Table S5, S6, S7)." - Table S7 shows comparison between Y13N and Y13, therefore, it is not relevant for this particlar sentence. 

lines 148-149 Although PF01556 is mentioned as not significant in comparison Z168N vsZ168 it is shown in Table S6 (lines 57 and 64). Why is PF00226 not shown n Table S6? 

line 177 - according to Figure 2a Z168N and Y13N have common 9197 DETs. Please, clarify.

lines 184-185 - in "ko:03110" and "ko:04141" colon should be removed

Discussian and Figure 5 - genes PB.28836.2, PB.27586.3, PB.45672.3, PB.25547.2, TraesCS3A02G034000.1, TraesCS5A02G547300.1 are not mentioned in the text and not discussed. As they are included in the group of 15 genes highly correlated with SSV they should be either mentioned in the text, or (what is better) all 15 genes should be organized in a separate Supplementary Table with some characterstics available from databases. This would be very useful for further research using association genetics and marker-assisted breeding. 

line 332-333 - "as the two cultivars may have 332 different HWM subunits." - it is rather easy to check if it is true not. Why not apply some simple PCR markers to verify this hypothesis? Even SDS-PAGE is not neccessary, I suppose. 

line 391 - "TraesCS4B02G222300" - it is not shown in Figure 5.

lines 401-402 - "Three randomly selected plots for each cultivar were treated twice with nitrogen fertilization" - what paricular fertilizer? What amount of N? What form of ions? Nitrates or ammonia? Side dressing or top dressing? 

line 404 - "Grains from the middle spikelet of a spike" - how many grain from how many plants?

The paper can be recommended for publication after improvement and addressing of the listed questions. 

Kind reagrds,

Reviewer.

Overall, the English is fine. However, to my mind, there are some corrections that should be made

line 43 - "human flavors" should be changed to "human tastes"

line 68 - "mensuration" should be changed to "measurment"

line 78  - please, intorduce ZM168 and YM13 as common bread wheat cultivars in the first sentence of this paragraph. 

line 207 - "GENA modules." - extra full stop (dot).

line 376 - PB.20810.1 should be in italics

In some subsection titles capital letters occur in the middle (e.g. 2.1, 2.6)

Reviewer 2 Report

The manuscript details the transcriptome of two wheat genotypes, with different quality attitude, in response to two nitrogen fertilization rates. Association with protein content and quality (SSV) was carried out, but no analysis on protein composition. The paper is well written and the aim clearly described.

A great value was attributed to heat shock proteins (HSP) in to regulating quality traits; however, these proteins are generally associated to abiotic stress events (heat); the contribution in forming gluten network could be limited, since their low amount. However, it is often associated better gluten quality (strength) to terminal heat during grain filling and ripeness, especially in temperate conditions that limit yield but improves quality (10.3390/plants10122599). The results of the current manuscript draw the attention to HSP in response to nitrogen for quality; this could be a promising target for future observations in order to make the regulation of gluten accumulation for quality more clear.

Finally, the manuscript is recommended to be acceptable for peer review journal after minor revision, following detailed.

Minor revisions:

N accumulation in grains is function of N supply (soil + fertilizer) and grain yield for dilution effect. Do the authors have data of grain yield in the different experimental conditions? Evaluating protein content itself could be limited without taking into account N accumulation dynamics. Indeed, the higher PC of Z168 could be due to a lower yield increase with N fertilization than Y13, with advantage to quality. We suggest to include this to have results more consistent.

lines 401: please detail nitrogen rate, source and timing of application
